# Electronic health record analysis identifies kidney disease as the leading risk factor for hospitalization in confirmed COVID-19 patients

**Matthew T. Oetjens**◉, **Jonathan Z. Luo**◉, **Alexander Chang**◉, **Joseph B. Leader, Dustin N. Hartzel, Bryn S. Moore, Natasha T. Strande**◉, **H. Lester Kirchner, David H. Ledbetter, Anne E. Justice, David J. Carey** *, **Tooraj Mirshahi**◉ *

Geisinger, Danville, Pennsylvania, United States of America

◉ These authors contributed equally to this work.
* tmirshahi@geisinger.edu (TM); djcarey@geisinger.edu (DJC)

**Data Availability Statement:** All relevant data are within the manuscript and its Supporting information files.

## Abstract

### Background

Empirical data on conditions that increase risk of coronavirus disease 2019 (COVID-19) progression are needed to identify high risk individuals. We performed a comprehensive quantitative assessment of pre-existing clinical phenotypes associated with COVID-19-related hospitalization.

### Methods

Phenome-wide association study (PheWAS) of SARS-CoV-2-positive patients from an integrated health system (Geisinger) with system-level outpatient/inpatient COVID-19 testing capacity and retrospective electronic health record (EHR) data to assess pre-COVID-19 pandemic clinical phenotypes associated with hospital admission (hospitalization).

### Results

Of 12,971 individuals tested for SARS-CoV-2 with sufficient pre-COVID-19 pandemic EHR data at Geisinger, 1604 were SARS-CoV-2 positive and 354 required hospitalization. We identified 21 clinical phenotypes in 5 disease categories meeting phenome-wide significance (P<1.60x10$^{-4}$), including: six kidney phenotypes, e.g. end stage renal disease or stage 5 CKD (OR = 11.07, p = 1.96x10$^{-8}$), six cardiovascular phenotypes, e.g. congestive heart failure (OR = 3.8, p = 3.24x10$^{-5}$), five respiratory phenotypes, e.g. chronic airway obstruction (OR = 2.54, p = 3.71x10$^{-5}$), and three metabolic phenotypes, e.g. type 2 diabetes (OR = 1.80, p = 7.51x10$^{-5}$). Additional analyses defining CKD based on estimated glomerular filtration rate, confirmed high risk of hospitalization associated with pre-existing stage 4 CKD (OR 2.90, 95% CI: 1.47, 5.74), stage 5 CKD/dialysis (OR 8.83, 95% CI: 2.76, 28.27), and kidney transplant (OR 14.98, 95% CI: 2.77, 80.8) but not stage 3 CKD (OR 1.03, 95% CI: 0.71, 1.48).

**Funding:** This work was supported by GM111913 from the NIH-NIGMS (NIH.GOV) to T.M. The funders had no role in study design, data collection and analysis, decision to publish, or preparation of the manuscript.

**Competing interests:** The authors have declared that no competing interests exist.

## Conclusions

This study provides quantitative estimates of the contribution of pre-existing clinical phenotypes to COVID-19 hospitalization and highlights kidney disorders as the strongest factors associated with hospitalization in an integrated US healthcare system.

## Introduction

Coronavirus disease 2019 (COVID-19) is an emerging illness caused by severe acute respiratory syndrome coronavirus 2 (SARS-CoV-2) infection. COVID-19 was declared a pandemic by the World Health Organization in March 2020. The United States reported the first case on January 22, 2020; by October 12[th], there were >7,740,000 total cases and >214,000 deaths (cdc.gov). The severity of COVID-19 illness is variable, ranging from asymptomatic [1] to severe complications that require hospitalization [2]. Several pre-existing conditions have been identified as risk factors for COVID-19-related hospitalization and death [3, 4]. A recent study developed a risk score that predicted progression to intensive care in hospitalized patients based on present and preexisting risk factors (e.g. chest radiographic abnormality, hemoptysis, dyspnea, history of cancer and other comorbidities) [5]. Comprehensive quantitative data on the contribution of pre-existing conditions to COVID-19 disease severity are still needed. We applied an agnostic cross-disease approach [6] to data captured in the patient's electronic health record (EHR) of SARS-COV-2-positive patients to identify associations between pre-existing conditions and COVID-19-related hospitalization.

## Methods

This study was conducted at Geisinger, an integrated health system in central and northeastern Pennsylvania [7]. This study was reviewed and approved by the Geisinger Institutional Review Board. This analysis includes patients with a laboratory confirmed diagnosis of COVID-19 reported between March 7, 2020 and May 19, 2020. All patients displayed symptoms that met CDC screening criteria for COVID-19 at the time of testing.

International Classification of Diseases Ninth (ICD-9) and Tenth (ICD-10) revision disease diagnosis codes and the last outpatient serum creatinine value were extracted from patients' EHR dated prior to January 1st, 2020. Potential risk factor phenotypes were defined by Phe-Codes mapped from ICD codes using PheCodes Map 1.2 [8] (https://phewascatalog.org/phecodes). For each individual, duplicate PheCode occurrences on the same date were dropped such that only one occurrence per date for a given PheCode remained. Cases for a phenotype were defined as having at least three occurrences of the PheCode; individuals with one or two occurrences were excluded from analysis of the phenotype, and the remaining individuals were classified as controls. To ensure that individuals in the study were adequately assessed for clinical history during clinical care, we restricted the analyses to individuals who were cases for at least one phenotype, which denotes that they have been clinically assessed on at least three distinct occasions. Our analysis required at least 20 cases and 20 controls for each phenotype among the 1,604 SARS-CoV-2 positive subjects, resulting in 313 distinct phenotypes. S1 Fig shows a flow diagram of the study design. The ICD code terminology used for the PheWAS data reflects the codes utilized by the PheCode Map 1.2 exactly.

We conducted additional analyses to further explore the relationship between kidney diseases and risk of COVID-19 hospitalization. We used estimated glomerular filtration rate

(eGFR), calculated by the CKD Epidemiology Collaboration equation, data up until August 2018 from the United States Renal Data System (USRDS [9] https://www.usrds.org/2018,), and ICD codes to categorize patients into 1 of 5 groups: 1) eGFR $\geq$ 60 ml/min/1.73m$^2$ without kidney transplant; 2) eGFR 30–59 ml/min/1.73m$^2$ without kidney transplant; 3) eGFR 15–29 ml/min/1.73m$^2$ without kidney transplant; 4) eGFR <15 ml/min/1.73m$^2$ or on dialysis; 5) kidney transplant with eGFR $\geq$ 15 ml/min/1.73m$^2$.

### Statistics

A phenome-wide association study (PheWAS) was performed to identify pre-existing conditions associated with hospitalization of patients with SARS-COV-2 infection. Tests were performed with Firth's logistic regression [8] adjusted for age, sex and race:

$$Hospitalization\ Status\ [Binary] \sim PheCode + Age + Sex + Race$$

Odds ratios (ORs) indicate the relative odds of COVID-19 related hospital admission given the presence of a pre-existing phenotype. We defined phenome-wide significance using a Bonferroni corrected p-value for the number of clinical PheCodes tested (p<0.05/313 = 1.60x10$^{-4}$).

### Results

Of 18,372 individuals tested for SARS-CoV-2 at Geisinger between March 7, 2020 and May 19, 2020; 15,707 tested negative, 2,665 tested positive, and 565 were admitted to the hospital. Among the total number tested, 12,971 met inclusion criteria for PheWAS analysis (Methods). Of the 12,971 SARS-CoV-2 tested patients used in PheWAS, 1,604 were positive for SARS-CoV-2 of whom 354 (22.1%) were admitted to the hospital (Table 1; demographics). Admitted patients were more likely to be older and male (p < 0.0001, Table 1). Of the 354 hospitalized patients, 106 were admitted to the ICU, 70 required ventilation, 71 died, and 54 remained hospitalized as of May 19, 2020.

We performed a PheWAS analysis to test for associations between COVID-19 related hospital admission and 313 clinical phenotypes (Fig 1; Table 2). Phenotypes that reached phenome-wide significance (p < 1.60x10$^{-4}$) fell into five disease categories: renal, cardiovascular,

**Table 1. Patient demographics and prevalence of select chronic conditions derived from EHR.**

| Inclusion Criteria | Population | SARS-CoV-2 tested | SARS-CoV-2 negative | SARS-CoV-2 positive | Admitted | p value |
|---|---|---|---|---|---|---|
| N | 1,069,142 | 12,971 | 11,367 | 1,250 | 354 | |
| Age, mean (SD) | 49.1 (25.4) | 49.2 (20.6) | 48.0 (20.2) | 54.8 (21.0) | 66.8 (17.3) | 3.88x10$^{-22}$ |
| Male, % | 45.9 | 37.1 | 36.8 | 36.4 | 50.3 | 3.22x10$^{-6}$ |
| BMI (SD) | 27.5 (8.2) | 30.2 (8.2) | 30.0 (8.3) | 30.8 (7.7) | 31.4 (8.1) | 0.229 |
| Current Smokers, % | 15.7 | 22.3 | 24.2 | 10.0 | 6.8 | 0.082 |
| Former Smokers, % | 24.9 | 32.4 | 31.9 | 32.7 | 47.2 | 7.73x10$^{-7}$ |
| Chronic kidney disease, % | 7.6 | 10.9 | 10.4 | 9.2 | 29.7 | 3.30x10$^{-19}$ |
| Chronic lung disease, % | 5.5 | 10.7 | 11.2 | 5.0 | 16.9 | 1.38x10$^{-13}$ |
| Diabetes mellitus, % | 16.1 | 23.1 | 22.3 | 23.9 | 45.5 | 4.11x10$^{-15}$ |
| Heart Failure, % | 5.1 | 7.4 | 7.2 | 5.5 | 17.8 | 2.66x10$^{-13}$ |
| Hypertension, % | 30.2 | 38.1 | 37.1 | 38.5 | 63.8 | 6.84x10$^{-14}$ |
| History Pneumonia, % | 3.5 | 9.6 | 9.4 | 5.8 | 27.7 | 3.49x10$^{-16}$ |
| Respiratory distress, % | 0.9 | 1.5 | 1.6 | 0.9 | 2.0 | 0.149 |

(p-values refer to comparison between SARS-CoV-2 positive and admitted patients, using unpaired t-test for Age and BMI, chi-square test for others. EHR Inclusion is defined in the results).

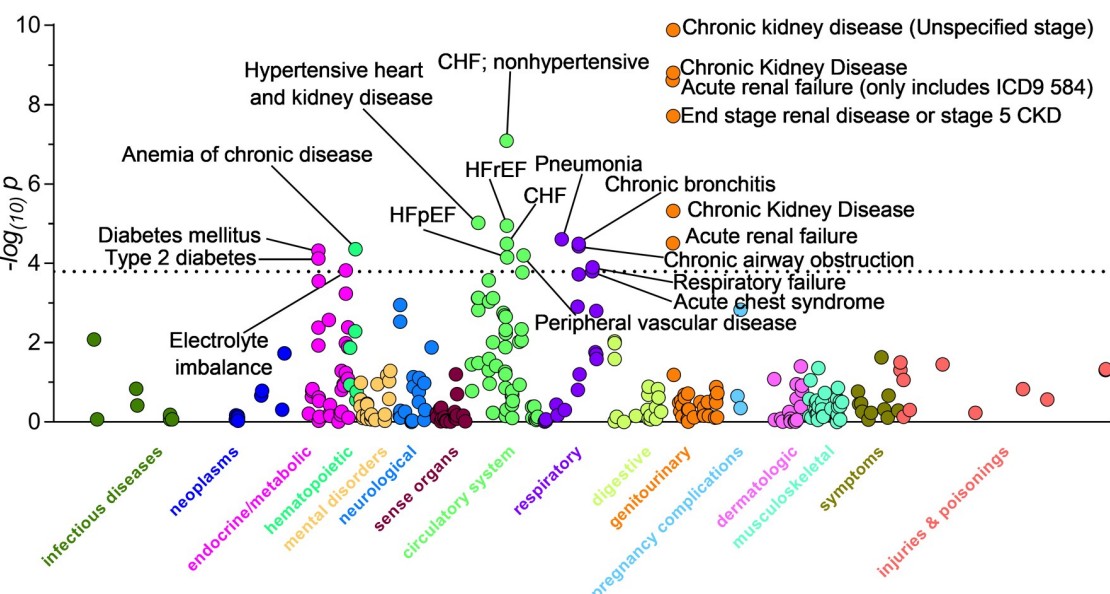

**Fig 1. Manhattan plot for clinical phenotypes associated with COVID-19 hospitalization.** Using a minimum case count of 20, we identified 313 clinical phenotypes, from PheCode Map 1.2, that could be used for these association studies. Dashed line denotes the Bonferroni significance ($1.60 \times 10^{-4}$).

**Table 2. Summary data for clinical phenotypes significantly associated with COVID-19 associated hospitalization adjusted for age, sex and race.**

| PheCode* | Description | OR | p | Cases |
|---|---|---|---|---|
| 585.3 | Chronic kidney disease (Unspecified stage) | 3.43 | $1.33 \times 10^{-10}$ | 172 |
| 585.34 | Chronic Kidney Disease, Stage IV | 11.85 | $1.59 \times 10^{-9}$ | 31 |
| 585 | Acute renal failure (only includes ICD9 584) | 2.95 | $2.43 \times 10^{-9}$ | 204 |
| 585.32 | End stage renal disease or stage 5 CKD | 11.07 | $1.96 \times 10^{-8}$ | 28 |
| 428 | Congestive heart failure; nonhypertensive | 3.35 | $8.13 \times 10^{-8}$ | 104 |
| 585.33 | Chronic Kidney Disease, Stage III | 2.68 | $4.74 \times 10^{-6}$ | 135 |
| 401.2 | Hypertensive heart and kidney disease | 2.99 | $9.54 \times 10^{-6}$ | 141 |
| 428.3 | Heart failure with reduced EF (HFrEF) | 4.82 | $1.13 \times 10^{-5}$ | 35 |
| 480 | Pneumonia | 3.17 | $2.48 \times 10^{-5}$ | 66 |
| 585.1 | Acute renal failure | 3.26 | $3.08 \times 10^{-5}$ | 63 |
| 428.1 | Congestive heart failure (CHF), unspecified | 3.8 | $3.24 \times 10^{-5}$ | 45 |
| 496.2 | Chronic bronchitis | 5.9 | $3.26 \times 10^{-5}$ | 24 |
| 496 | Chronic airway obstruction, unspecified | 2.54 | $3.71 \times 10^{-5}$ | 101 |
| 285.2 | Anemia of chronic disease | 4.86 | $4.36 \times 10^{-5}$ | 30 |
| 250 | Diabetes mellitus | 1.83 | $4.83 \times 10^{-5}$ | 341 |
| 443.9 | Peripheral vascular disease | 3.25 | $6.37 \times 10^{-5}$ | 53 |
| 428.4 | Heart failure with preserved EF (HFpEF) | 3.26 | $7.01 \times 10^{-5}$ | 56 |
| 250.2 | Type 2 diabetes | 1.8 | $7.51 \times 10^{-5}$ | 336 |
| 509.1 | Respiratory failure | 4.11 | $1.26 \times 10^{-4}$ | 33 |
| 276.1 | Electrolyte imbalance not elsewhere classified | 2.69 | $1.53 \times 10^{-4}$ | 76 |
| 509 | Acute chest syndrome | 3.69 | $1.59 \times 10^{-4}$ | 38 |

*Each PheCode represents at least 1 and often several ICD-9 or ICD-10 codes. For list of ICD-9 codes included in each PheCode: https://phewascatalog.org/phecodes.
For list of ICD-10 codes included in each PheCode: https://phewascatalog.org/phecodes_icd10cm.

endocrine/metabolic, respiratory, and hematopoietic. The most significant associations (smallest p value and largest OR) were related to disorders of renal function, including chronic kidney disease (unspecified stage) (OR = 3.43, 95% CI [2.36,5], p = 1.33 x $10^{-10}$), end stage renal disease or stage 5 CKD (OR = 11.07, 95% CI [4.54,26.97], p = 1.96 x $10^{-8}$), stage III chronic kidney disease, (OR = 2.68, 95% CI [1.76,4.06], p = 4.74 x $10^{-6}$) and acute renal failure (OR = 3.26, 95% CI [1.89,5.62], p = 3.08 x $10^{-5}$). Six disorders in the cardiovascular disease category, including nonhypertensive congestive heart failure (OR = 3.35, 95% CI [2.16,5.2], p = 8.13 x $10^{-8}$), and peripheral vascular disease (OR = 3.25, 95% CI [1.84,5.71], p = 6.37 x $10^{-5}$) reached phenome-wide significance. Type 2 diabetes (OR = 1.8, 95% CI [1.35,2.41], p = 7.51 x $10^{-5}$) was among three disorders in the endocrine/metabolic disease category that reached phenome-wide significance. Within the respiratory disease category, 5 conditions were significant, including chronic airway obstruction (OR = 2.54, 95% CI [1.65,3.93], p = 3.71 x $10^{-5}$), pneumonia (OR = 3.17, 95% CI [1.89,5.33], p = 2.48 x $10^{-5}$), and chronic bronchitis (OR = 5.9, 95% CI [2.58,13.48], p = 3.26 x $10^{-5}$). Lastly, we identified a single hematopoietic association with anemia of chronic disease (OR = 4.86, 95% CI [2.33,10.15], p = 4.36 x $10^{-5}$). A list of all 313 conditions tested in PheWAS is shown in S1 Table.

In addition to PheWAS findings, we also used several algorithms that have been extensively validated to define disease phenotypes using EHR data (Table 1, S1 Methods). We observed significantly higher frequencies of several of these phenotypes in hospitalized patients; chronic kidney disease was most strongly associated with hospitalization (S2 Fig). In analyses using eGFR data and USRDS data, stage 4 CKD (OR 2.90, 95% CI: 1.47, 5.74), and stage 5 CKD/dialysis (OR 8.83, 95% CI: 2.76m 28.27) were associated with increased risk of COVID-19 hospitalization whereas stage 3 CKD was not (OR 1.03, 95% CI: 0.71, 1.48). Five (71%) out of 7 patients with history of kidney transplant were hospitalized (OR 14.98, 95% CI: 2.77, 80.88). Among 565 hospitalized patients, which included those who were not included in the PheWAS due to limited EHR data, 122 had some history of CKD. The metabolic burden among these 122 patients was higher than those without CKD. These patients were typically older and had significantly higher rates of death (Table 3).

## Discussion

The outbreak of COVID-19 has spurred unprecedented efforts to characterize biological and clinical aspects of the disease [10, 11]. COVID-19 is the first pandemic in the digital health age, which has allowed rapid epidemiologic studies [1, 13]. Data from government agencies such as Centers for Medicare & Medicaid Services (cms.gov/covid-19-data-snapshot-fact-sheet), encompass large cohorts but are mostly snapshots that lack granular data and rely heavily on claims and provider supplied data. Here, we used data from an integrated health system with outpatient and inpatient COVID-19 testing capacity and utilized a PheWAS study design to conduct a comprehensive analysis of clinical phenotypes associated with increased risk of COVID-19 related hospital admission. To control for potential bias related to exposure to the SARS-CoV-2 virus, we limited our study population to SARS-CoV-2 positive patients screened at Geisinger. Additional analyses using eGFR and USRDS data confirmed our findings that patients with stage 4–5 CKD, ESRD on dialysis or with kidney transplant are at extremely high risk for severe complications due to COVID-19 (Table 4). These findings complement findings from the OPEN Safely study, which found similar results but was limited by including clinically suspected (non laboratory confirmed) COVID-19 [12]. As well as the CMS reports showing higher risk of hospitalization among ESKD patients (not adjusted for covariates). Our finding of high risk of hospitalization in kidney transplant patients mirrors that of a case series of 36 consecutive kidney transplant patients at Montefiore where 28/36 (78%) were

**Table 3. Clinical outcomes for hospitalized patients with known history of chronic kidney disease.**

| | CKD | non-CKD |
|---|---|---|
| Patient Count | 122 | 443 |
| Age (±SD) | 74.8±11.67 | 61.75±17.67 |
| BMI (±SD) | 29.3±7.5 | 31.0±8.6 |
| Diabetes | 50.82% (62/122) | 33.41% (148/443) |
| Hypertension | 76.23% (93/122) | 41.08% (182/443) |
| Heart Failure | 40.16% (48/122) | 6.55% (29/443) |
| Chronic Lung Disease | 27.87% (34/122) | 8.13% (36/443) |
| Ever Smoker | 59.02% (72/122) | 41.99% (186/443) |
| Days in Hospital (±SD)) | 7.5±5.9 | 7.4±6.6 |
| Admitted to ICU | 27.8% (34/122) | 30.7% (136/443) |
| Days in ICU (±SD) | 7.5±8.6 | 7.6±7.0 |
| Ventilator used | 16.4% (20/122) | 20.3% (90/443) |
| Days on Ventilator (±SD) | 10.0±8.70 | 8.3±6.9 |
| Min Resp Rate (±SD) | 13.2±3.9 | 13.5±4.2 |
| SPO2 at admission (±SD) | 95.0±6.4 | 93.9±6.5 |
| AVG SPO2 (±SD) | 94.8±4.0 | 95.3±2.3 |
| Died | 25.4% (31/122) | 13.3% (59/443) |

Among all 565 COVID-19 patients admitted to hospital, 122 had a history of CKD. Patients with known history of CKD were older, typically had higher disease burden and higher death rate (Odds Ratio = 2.2; p = 0.002, Fisher's exact test).

hospitalized [13]. The findings reported here identify co-morbidities that impact the clinical course of COVID-19 and may be used to identify individuals at greatest risk for COVID-19-related complications.

The majority of conditions associated with increased risk of COVID-19 related hospital admission have been suggested in previous studies, including diabetes, heart failure, hypertension, and chronic kidney disease [14–16]. What is striking from our results is the magnitude of the kidney disease-related risk. Patients with end-stage renal disease were at 11-fold increased odds of hospitalization (Table 2). How clinical conditions increase the risk of COVID-19-related complications is not fully clear yet. The physiological stress caused by excessive inflammatory response to SARS-COV-2 infection could destabilize organs already weakened by chronic disease [17]. Alternatively, direct organ-specific injury from SARS-CoV-2 infection

**Table 4. Association between CKD phenotypes and COVID-19 associated hospitalization.**

| | Unadjusted | P value | Adj. for age, sex, race/ethnicity | P value |
|---|---|---|---|---|
| eGFR $\geq$ 60 (n = 1087) | Ref | | Ref | |
| eGFR 30–59 (n = 246) | 1.76 (1.28, 2.42) | <0.001 | 1.03 (0.71, 1.48) | 0.88 |
| eGFR 15–30 (n = 38) | 5.22 (2.73, 9.99) | <0.001 | 2.90 (1.47, 5.74) | 0.002 |
| eGFR <15 or on dialysis (n = 16) | 13.43 (4.29, 42.08) | <0.001 | 8.83 (2.76, 28.27) | <0.001 |
| Kidney transplant (n = 7) | 11.36 (2.19, 59.00) | 0.004 | 14.98 (2.77, 80.88) | 0.002 |

Participants were classified into 1 of 5 groups based on their last eGFR before 1/1/2020, USRDS data, and ICD codes. Kidney transplant patients were classified separately and had a range of eGFR from 19.84 to 95.25 ml/min/1.73m$^2$. N = 1394 for this analysis as eGFR was available for 1393 individuals; 1 patient with history of kidney transplant with no eGFR values before 1/1/2020 was included in this analysis.

could act as a "second-hit" to these organs. Consistent with this hypothesis, kidney and heart are among the tissues with the highest expression of *ACE2*, a SARS-CoV-2 receptor [18].

The current study has several limitations. The sample size is relatively small, and the available data are limited to information captured in the EHR. Nevertheless, we were able to identify several highly significant traits associated with hospitalization, many of which are consistent with previous reports. The study population is subject to potential bias resulting from the availability of testing, which largely excluded asymptomatic individuals and an enrichment of individuals from nursing homes and healthcare workers [19]. To partially overcome this, we included in our analyses individuals who were tested in a single health system. This population is predominantly Caucasian, which may limit the generalization of our findings to other racial and ethnic groups. A recent study indicated that hospitalization rates may differ between racial and ethnic groups with COVID-19 [20].

In conclusion, this study leverages extensive longitudinal EHR data prior to the COVID-19 pandemic to identify pre-existing clinical phenotypes associated with increased risk of COVID-19 hospitalization. These results provide key information for public policymakers highlighting the need to prevent COVID-19 related illness in patients with kidney disease and other high-risk conditions.

## Supporting information

**S1 Table. A list of all 313 conditions tested in PheWAS.**
(XLSX)

**S1 Methods.**
(PDF)

**S1 Fig. Study flow diagram.**
(PDF)

**S2 Fig. Prevalence of validated disease phenotypes using EHR data among the total EHR population, all those tested for COVID-19, those who tested negative for COVID-19, COVID-19(+) individuals not needing admission and hospitalized for COVID-19(+) individuals.**
(PDF)

## Acknowledgments

Data reported here have been supplied by the United States Renal Data System (USRDS). The interpretation and reporting of these data are the responsibility of the author(s) and in no way should be seen as an official policy or interpretation of the U.S. government.

## Author Contributions

**Conceptualization:** Matthew T. Oetjens, Jonathan Z. Luo, Alexander Chang, Anne E. Justice, David J. Carey, Tooraj Mirshahi.

**Data curation:** Jonathan Z. Luo, Joseph B. Leader, Dustin N. Hartzel, Tooraj Mirshahi.

**Formal analysis:** Matthew T. Oetjens, Jonathan Z. Luo, Alexander Chang, Bryn S. Moore, Tooraj Mirshahi.

**Methodology:** Jonathan Z. Luo.

**Project administration:** Tooraj Mirshahi.

**Resources:** David H. Ledbetter, David J. Carey, Tooraj Mirshahi.

**Supervision:** David J. Carey, Tooraj Mirshahi.

**Validation:** Joseph B. Leader, Dustin N. Hartzel, Bryn S. Moore, Natasha T. Strande, H. Lester Kirchner, David H. Ledbetter, Anne E. Justice.

**Visualization:** Tooraj Mirshahi.

**Writing – original draft:** Matthew T. Oetjens, David J. Carey, Tooraj Mirshahi.

**Writing – review & editing:** Matthew T. Oetjens, Jonathan Z. Luo, Alexander Chang, Joseph B. Leader, Bryn S. Moore, Natasha T. Strande, H. Lester Kirchner, David H. Ledbetter, Anne E. Justice, David J. Carey, Tooraj Mirshahi.

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
