## [Decision Letter · Decision Letter 0]

11 Sep 2020

PONE-D-20-26257

Electronic health record analysis identifies kidney disease as the leading risk factor for hospitalization in confirmed COVID-19 patients.

PLOS ONE

Dear Dr. Mirshahi,

Thank you for submitting your manuscript to PLOS ONE. After careful consideration, we feel that it has merit but does not fully meet PLOS ONE’s publication criteria as it currently stands. Therefore, we invite you to submit a revised version of the manuscript that addresses the points raised during the review process.

Both reviewers found substantial merit in the paper, but listed multiple issues that require attention. I agree with their comments and hope you can revise the paper according to their suggetions.

We look forward to receiving your revised manuscript.

Kind regards,

Harald Mischak

Academic Editor

PLOS ONE

Journal Requirements:

2. Please state whether the baseline demographic characteristics of the study populations were recorded. If so, please provide a table summarising these.

Reviewers' comments:

Reviewer's Responses to Questions

**Comments to the Author**

1. Is the manuscript technically sound, and do the data support the conclusions?

Reviewer #1: Yes

Reviewer #2: Yes

2. Has the statistical analysis been performed appropriately and rigorously? 

Reviewer #1: Yes

Reviewer #2: Yes

3. Have the authors made all data underlying the findings in their manuscript fully available?

Reviewer #1: No

Reviewer #2: Yes

4. Is the manuscript presented in an intelligible fashion and written in standard English?

Reviewer #1: Yes

Reviewer #2: Yes

5. Review Comments to the Author

Reviewer #1: Authors pursue a very interesting and commendable endeavor of PheWAS of CoVID-19 positive patients to identify those at high risk for CoVID-related hospitalization. Following are our comments associated with each section:

Introduction

Line 6 regarding ‘Case reports have shown that older age, hypertension and diabetes are risk factors for COVID-19-related complications’. Would suggest omitting this statement (case reports) since there are a plethora of studies now available that demonstrate risk factors associated with CoVID-19 related complications. Would suggesting at least 2-3 major studies. For example, the study you quoted as reference [10].

Results

1. Second paragraph- would replace ‘acute renal failure’, with ‘acute kidney injury’ as terminology. Would be prudent to separate out pre-existing and new acute condition, since the intent is to find risk factors for CoVID related hospitalizations.

2. Also is the AKI at the time of diagnosis of CoVID-19 and how have you defined an AKI? The Methods specify ‘last outpatient serum creatinine value’, which could be very variable since most patients may not have had a creatinine checked proximal to diagnosis of CoVID-19. As per most recent data, AKIs in context of CoVID-19 have only developed during hospitalization course, rather than at the time of diagnosis. Please clarify your definition of AKI.

3. Second paragraph- how did you define ESKD as well, since you have mentioned CKD stage 5 and ESKD together. Please clarify.

4. Second paragraph- please explain what encompasses the term ‘nonhypertensive congestive heart failure’. A slightly non-descript term and might be confusing to the reader.

5. Second paragraph- you have listed pneumonia as a risk factor. Again, an acute event and is this also present at the time of diagnosis of CoVID-19.

6. On page 4 of the manuscript, there is a typo for number 15 following eGFR 15-29: 3) eGFR 15-29 15 ml/min/1.73m2 without kidney transplant;

Discussion

1. First paragraph- ‘However, initial reports of COVID-19 describing potential risk factors for hospitalization and other adverse outcomes have often been derived from cohorts without sufficient pre-hospitalization data that is often incomplete or missing’. Unfortunately, this is now an inaccurate statement- you could say that in March or April 2020. But as I quoted an example in the introduction section, there a good few studies now available looking at hospitalization risk and adverse outcome. Here’s another large study of >300,000 in PNAS (https://www.pnas.org/content/early/2020/08/10/2011086117). More- https://academic.oup.com/cid/advance-article/doi/10.1093/cid/ciaa1012/5872581,

https://www.ncbi.nlm.nih.gov/pmc/articles/PMC7245300/

Please consider updating the manuscript. Your study certainly does add to the literature of risk factors, and you could elucidate that in the discussion section.

2. First paragraph- ‘Further, confounders such as age, sex, and race are often not accounted for’. Again, would be inaccurate to state that as well. Please consider omitting, based on examples discussed above. Another example- https://bmcinfectdis.biomedcentral.com/articles/10.1186/s12879-020-05144-x

https://www.nejm.org/doi/full/10.1056/NEJMsa2011686

3. First paragraph- ‘As well as the CMS reports showing higher risk of hospitalization among ESKD patients (not adjusted for covariates). Our finding of high risk of hospitalization in kidney transplant patients mirrors that of a case series of 36 consecutive kidney transplant patients at Montefiore where 28/36 (78%) were hospitalized’,

Again, multiple studies have since been been published regarding CoVID-19 in ESRD/ESKD patients and kidney transplant recipients.

Would consider discussing studies such as https://www.kidney-international.org/article/S0085-2538(20)30945-5/fulltext, for ESKD in addition to CMS reports.

Here are bigger studies on kidney transplant recipients for the discussion. The Montefiore study is small and is from early days of COVID-19- https://www.kidney-international.org/article/S0085-2538(20)30961-3/fulltext#.X0OiqBDfdDg.twitter

https://onlinelibrary.wiley.com/doi/full/10.1111/ajt.16185

Again, consider updating the manuscript with these more recent studies.

4. First paragraph- ‘The findings reported here identify pathophysiologies that impact the clinical course of COVID-19 and may be used to identify individuals at greatest risk for COVID-19-related complications’. The term ‘pathophysiologies’ seems misleading – consider using ‘co-morbidities’ instead.

5. Second paragraph- ‘How clinical conditions increase the risk of COVID-19-related complications is not known’. Again, as discussed before. Not necessarily an accurate statement as knowledge has quickly evolved. Consider updating or omitting.

6. Third paragraph- ‘Our findings also have implications for studies that seek to find genetic variants that alter the course of the disease (e.g. https://www.covid19hg.org/). We suspect that some genetic variants will be associated with these traits based on their effects on pre-existing clinical conditions’. We are not certain how this statement regarding genetic studies fits in the discussion. Consider clarifying or omitting.

7. We tend to agree with the authors that one of the most important limitations of their study is generalization as their population is mostly Caucasian. Also, consider adding race to table 1 of participants characteristics.

We hope these few comments will help the authors improve on their interesting manuscript and we thank the editor for entrusting our opinion.

Reviewer #2: This is an interesting study adding important information on the importance of chronic kidney disease as risk factor for COVID-19 associated hospitalization rate in the USA.

Comments:

Of high interest would be the outcome of the 354 patients admitted to the hospital. It would be very interesting to provide data on the outcome of those patients after hospitalisation, divided according to the different diseases/phenotypes (outcome: admission to ICU, need for ventilation, fatal outcome)? Did CKD patients (e.g. dialysis patients or stage IV and V CKD patients) have a higher risk for disease progression or death?

The high OR of 11 for hospitalisation with COVID-19 for patients with stage 5 CKD and/or ESKD is astonishing, especially when compared to other phenotypes, such as CHF. This point was not really adressed in the discussion. The renal tropism of SARS-CoV-2 and the high ACE2R expression in the kidneys are probably not part of the explanation. It would be interesting to have some discussion on this.

Additionally, what triggered the hospitalisation for those patients with COVID-19? Could it be that patients with ESKD were more likely to be hospitalized for fear of progression to a more serious disease and perhaps because of a strong interest in isolating these patients from other dialysis patients and keeping them away from dialysis units rather than because of the severity of COVID-19 disease? Since hospitalization was the endpoint of this study, there should be some comments on the reasons/triggers for hospitalization.

Table 1:

As reported in the text, there were 1,604 persons tested positive for SARS-CoV-2. But in table 1 only 1,250 positive patients are reported. Please correct.

Please explain the meaning of the numbers in the first line (inclusion Criteria, N (%)). Percent of what? Of 18,372 persons with testing for SARS-CoV-2 positivity, 12,971 fullfilled inclusion criteria of this study. This is 70.6% of all tested persons. The number of persons with fullfilling inclusion criteria and a negative SARS-CoV-2 Test were 11,367. But what does 72,4% refer to? It should be either 61,9% of the tested population or 87,6% of the tested population with inclusion criteria. Please, re-check your calculations and numbers of this table, especially line 1 and please explain.

Table 2:

Almost 50% of all CKD cases are „unspecified“ regarding to CKD stage (line 1). Since CKD stage seems to have tremendous impact on the COVID-19 ass. hospitalization risk, it would be most welcome if those 172 „unspecified stage “ patients could be re-classified according to eGFR data and added to each staged CKD phenotypes? In table 3 it is said, that eGFR data are available for 1393 individuals…

The small number of patients with stage IV and V/VD CKD lowers the power of this analysis. It does not seem biologically plausible that the Odds for hospitalisation with COVID-19 increases from 2,68 with CKD III to 11,85 with CKD IV.

In table 3 the numbers are much more plausible with the association of COVID-19 related hospitalization with eGFR stages. Stage III had an OR of (adj) 1,03, stage IV 2,90 and stage V 8,83. In table 3 unadjusted and adjusted OR were given. Please confirm that ORs in table 2 are adjusted values (should be also stated in the table legend)

Figure 1:

please explain why there are three (!) circles for chronic kindey diseases at different risk points in the graph (and two circels for acute renal faliure as well). This is confusing and should be clarified.

Further comments:

In the discussion the two terms "ESRD" and "ESKD" are used. This should be harmonized.

(literature suggestion on nomenclature in kidney diseases: NDT, Vol 35, 2020, 1077–1084, https://doi.org/10.1093/ndt/gfaa153

I suggest not to use the term „COVID-19 positive patients“, neither in the txt nor in table one. It should be either „SARS-CoV-2 positive“ or „SARS-CoV-2 infected patients“ or „patients with COVID-19“. ´COVID-19 is not a virus (SARS-CoV-2 is…), it’s a disease.

6. PLOS authors have the option to publish the peer review history of their article (what does this mean?). If published, this will include your full peer review and any attached files.

Reviewer #1: No

Reviewer #2: No

---

## [Author Response · Author response to Decision Letter 0]

12 Oct 2020

We would like to thank the two reviewers for the insightful, and positive review of our manuscript. We have addressed all the points raised and edited the manuscript to reflect the changes as suggested by reviewers. Point by point response to each of the questions is denoted in red in the attached file..

Sincerely, 

Tooraj Mirshahi

---

## [Decision Letter · Decision Letter 1]

27 Oct 2020

PONE-D-20-26257R1

Electronic health record analysis identifies kidney disease as the leading risk factor for hospitalization in confirmed COVID-19 patients.

PLOS ONE

Dear Dr. Mirshahi,

Thank you for submitting your manuscript to PLOS ONE. After careful consideration, we feel that it has merit but does not fully meet PLOS ONE’s publication criteria as it currently stands. Therefore, we invite you to submit a revised version of the manuscript that addresses the points raised during the review process.

We look forward to receiving your revised manuscript.

Kind regards,

Harald Mischak

Academic Editor

PLOS ONE

Additional Editor Comments (if provided):

Theer are a few minor issues to be corrected, please change the paper accordingly and resubmit, so it can be accepted.

Reviewers' comments:

Reviewer's Responses to Questions

**Comments to the Author**

1. If the authors have adequately addressed your comments raised in a previous round of review and you feel that this manuscript is now acceptable for publication, you may indicate that here to bypass the “Comments to the Author” section, enter your conflict of interest statement in the “Confidential to Editor” section, and submit your "Accept" recommendation.

Reviewer #1: All comments have been addressed

Reviewer #2: All comments have been addressed

2. Is the manuscript technically sound, and do the data support the conclusions?

Reviewer #1: Yes

Reviewer #2: Yes

3. Has the statistical analysis been performed appropriately and rigorously? 

Reviewer #1: N/A

Reviewer #2: Yes

4. Have the authors made all data underlying the findings in their manuscript fully available?

Reviewer #1: Yes

Reviewer #2: Yes

5. Is the manuscript presented in an intelligible fashion and written in standard English?

Reviewer #1: Yes

Reviewer #2: Yes

6. Review Comments to the Author

Reviewer #1: Thank you for taking into account the comments of the reviewers in updating your manuscript. I have no further comments at this time.

Reviewer #2: I thank the authors for their efforts to adress our points raised..

Even though some of the points discussed in response to our comments could be part of an interesting debate I feel the paper is now improved and within the discussed limitations this paper really adds important information and insights to the current knowledge of risk factors for COVID-19 hospitalisations. I recommend it for publishing in PLOS ONE after very few very minor revisions:

a) Citation number 10 is cited twice in the first sentence of the discussion. Please correct.

b) Second sentence of the discussion: citation has sliped behind the sentence point

c) Citation number 10 is now incorrect. The current citation refers to an non-peer-reviewed draft of the article on a pre-print-server. The peer-reviewed final paper is published here: Brain Behav Immun. 2020 Jul; 87: 184–187. Published online 2020 May 23. doi: 10.1016/j.bbi.2020.05.059 PMCID: PMC7245300 PMID: 32454138

It has a new title as well: Lifestyle risk factors, inflammatory mechanisms, and COVID-19 hospitalization: A community-based cohort study of 387,109 adults in UK

d) there are a few more minor citation issues (e.g. citation 11 has blank spaces inbetween USA (U S A) or citation 14 - 16 should be combined into the same brackets), which will probably be adressed by the ediorial office before final proof.

7. PLOS authors have the option to publish the peer review history of their article (what does this mean?). If published, this will include your full peer review and any attached files.

Reviewer #1: No

Reviewer #2: **Yes: **Ralph Wendt

---

## [Author Response · Author response to Decision Letter 1]

27 Oct 2020

Reviewers #2, Thanks for your keen eye for the details. As requested we updated ref#10, to the published version of that paper (the preveious citation was for the same paper in medRxiv).

We have reworked the citation punctuation for references #10-11 as well as 14-16, all in the discussion.

---

## [Editor Report · Decision Letter 2]

29 Oct 2020

Electronic health record analysis identifies kidney disease as the leading risk factor for hospitalization in confirmed COVID-19 patients.

PONE-D-20-26257R2

Dear Dr. Mirshahi,

We’re pleased to inform you that your manuscript has been judged scientifically suitable for publication and will be formally accepted for publication once it meets all outstanding technical requirements.

Kind regards,

Harald Mischak

Academic Editor

PLOS ONE
---

## [Editor Report · Acceptance letter]

4 Nov 2020

PONE-D-20-26257R2 

Electronic health record analysis identifies kidney disease as the leading risk factor for hospitalization in confirmed COVID-19 patients 

Dear Dr. Mirshahi:

I'm pleased to inform you that your manuscript has been deemed suitable for publication in PLOS ONE. Congratulations! Your manuscript is now with our production department. 

Kind regards, 

on behalf of

Prof. Harald Mischak 

Academic Editor

PLOS ONE